# Effectiveness of Electrical and Optical Detection at Pixel Circuit on Thin-Film Transistors

**DOI:** 10.3390/mi12020135

**Published:** 2021-01-27

**Authors:** Fu-Ming Tzu

**Affiliations:** Department of Marine Engineering, National Kaohsiung University of Science and Technology, Kaohsiung 80543, Taiwan; fuming88@nkust.edu.tw

**Keywords:** open and short defect, pixel circuit, thin-film transistor, automatic optical inspection

## Abstract

The paper presents a typology of electrical open and short defects on thin-film transistors (TFT) using an electrical tester and automatic optical inspection (AOI). The experiment takes the glass 8.5th generation to detect the electrical characteristics engaged with time delay and integration (TDI) charged-coupled-devices (CCDs), a fast line-scan, and a review CCD with five sets of magnification lenses for further inspection. An automatic data acquisition program (ADAP) controls the open/short (O/S) sensor, TDI-CCD, and motor device for machine vision and statistics of substrate defects simultaneously. Furthermore, the quartz mask installed on AOI verified its optical resolution; a TDI-CCD can grab an image of a moving object during transfers of the charge in synchronous scanning with the object that is significant.

## 1. Introduction

A thin-film-transistor (TFT) is an active matrix pixel circuit to switch the liquid crystal rotation between the color filter (CF) and TFT that is liquid crystal display (LCD) [1,2,3]. A thousand million pixels are distributed onto the TFT panel as the electrical switch is manufactured as one of the types of field-effect transistors [4,5,6]. The production method deposits various thin films on the substrate, such as active semiconductor layers, dielectrics, and metal electrodes, which deposits a thin film on the substrate. In the study, the TFT was fabricated by the back-channel etching five-layer processes, including gate metal, dielectric deposition, source/drain metal, silicon nitride passivation, and indium tin oxide (ITO) electrode. Significantly, the metal layer such as gate, source, and drain are critical layers for electrical delivery.

In contrast, the signal transmission is a challenge if any foreign particle adheres to the metal line. Thus, the paper investigated whether the defect characteristics are based on open, short, and cross short defects on the thin-film transistor pixel circuit. On the other hand, the TFT resolution moves forward to the high-definition [7,8,9] as the market requests a wide view illumination coupled to the ultra-high definition. The pixel circuit in TFT is trending toward smaller and smaller dimensions. Typically, the active pixel resolution of the extended graphics array (XGA) is designated for 100 × 300 μm. The pixel circuit with various resolutions in TFT is dependent on the product requirement; however, the resolution is even smaller beyond, there is less resolution, larger pixel size, and lower image quality, and vice versa. Moreover, the refereed electrical voltage 5 V is for the working potential energy on the pixel circuit.

With the portable device demand blooming, electronic products’ development is toward properties of the light body, slim think, short volume, smaller size and more saturation color, various hue, and more color space high resolution and wide range chromaticity. However, the vision of the human being is quite sharp to perceive the image. Thus, each defect on the screen is a notorious yield killer. These kinds of defects typically are open, short on the gate, source, and drain related to electrical transportation, which decreases the signal delivery as stumble product. Once the defect includes open, short, and cross-type adheres on TFT, the electrical delivery incurs a black spot and color nonuniformity to appear on the screen. 

Jacobs et al. [10] utilized the laser-based fault isolation methodologies to identify the defect’s localization for open and short at 1 × 5 μm via-last through-silicon via (TSV) structures on 3-dimensional system-on-chip. A light-induced capacitance alteration (LICA) was measured on an open defect of a 1 × 5 μm TSV chain structure with a manufacturing fault. As a result, the open-type and short-type defect measured the cross-sectioning on the chip by scanning electron microscope (SEM) and focused-ion beam (FIB). Li and Li [11] proposed a defect detection method of bare printed circuit boards (PCB). They found proposed algorithm could accurately detect typical defects of the bare PCB for short circuit, open circuit, scratches, and voids. Lin et al. [12] measured the width and gap of etching transistors using sub-pixel accuracy estimation in TFT-LCD panels. The experiment took an automatic measure device to inspect TFT-LCD testing pattern defect such as open and short, marking, spot particle, or scratching on the panel caused by over-etching. The paper developed an automatic testing TFT-LCD etching system on stability, efficiency, and effectiveness. Yang et al. [13] presented an automatic optical inspection (AOI) system for detection of thin-film transistor (TFT) array defects using gray level co-occurrence matrix (GLCM) and MATLAB region props function for 53 defect panel of TFT array. The authors put the data into the neural network to train the defect classifier, and the result improved the testing efficiency and reduced the manufacturing costs. Chen and Kuo [14] detected the panel defect for TFT-LCD using automatic inspection on the defect mura through the discrete cosine transform (DCT) principle and background image filtering strategy. As a result, the significant level of mura defects is quantified by just noticeable difference (JND) [15]. 

The above literature reviews the articled related open/short defect as independent, i.e., either the electrical tester or AOI. In contrast, they do not further investigate a methodology to engage both open/short and optical inspection simultaneously. Thus, the paper proposes an integrated method for open/short and a fast line-scan to check the TFT panel defect. Figure 1 illustrates the topology of the open/short defect on TFT, in which Figure 1A is the normal condition, that the electrical conductivity is at normal condition. Figure 1B shows the open defect to block the electrical connection as a red circle indicating poor signal delivery, and Figure 1C is the short defect across the several metal lines as the red circle.

## 2. Principles of the Automatic Optical Inspection for TFT

The image perception is quite critical in the display as human vision is so sensitive. Thus, image quality needs to be more precious than just noticeable difference (JND) [16,17]. However, human vision does not have only scientifically quantitative properties but also objective properties. Whereas, the automatic optical inspection can play the role of providing accurate quantitative data that outperforms the objective visual. The AOI judges the substrate to conform to the standard device, which takes the gray-level distribution by line-scan and measures the color difference. The image quality of AOI is dependent on the resolution and optical detectability.

Moreover, the AOI can wildly apply in the industry, semiconductor, display, lighting, solar panel, motor mobile, aviation, and shipping, etc. Furthermore, AOI utilizes an optical imaging method consisting of a photosensor, reflective lens, blue LED light source [18], electrical cable, frame grabber, and data processor to simulate the visual imaging function of the human eye. The image is sent to the computer process system to replace the objective human judgment that performs data processing. Finally, it feedbacks the results to the server. 

Within the photosensor industry, time delay and integration (TDI) is a scanning device that increases the sensitivity of the multiple line-scan [17]. TDI-CCD has an area array structure but has a linear array. Compared with ordinary linear array CCDs, it has the function of multi-stage delay integration. Multiple linear arrays are arranged in parallel from its structure, and the pixels are arranged in a rectangular shape in the linear array. TDI-CCD accumulation is charged along the moving direction drives the accumulated stage from the 1st to the 96th in the range. During the imaging process, with camera movement, the TDI-CCD is illuminated sequentially, and the charge is gradually accumulated. As a result, the charge packets accumulate through multiple delay integration are transferred to readout in the register and transmitted from the first stage through the operational amplifier. It can be seen from the electrical performance that TDI-CCD is unidirectional driven imaging. Compared with general line-scan CCDs, TDI utilizes variable integration levels of 6, 12, 24, 48, 96 to increase the exposure time. When the sensor is imaging because the signal storage is proportional to the exposure time, TDI-CCD increases the collected photons by extending the exposure time. It has higher sensitivity than the general linear CCD and takes imaging in low-light environments, whatever the scanning speed. TDI-CCD outperformance obtains high sensitivity without sacrificing spatial resolution, making it have a wide range of application prospects in high-speed and low-light fields [19,20]. 

The TDI-CCD applied a popular commercial-off-the-shelf type among the photosensors. It is the fastest progressive line scan, manufactured by Teledyne DALSA, model product H.S. 8K TDI CCD, a specification of Piranha HS 8K engaged the scanning rate 68kHz [17]. Thus, the photosensor indicates a quick responsivity than other traditional line CCD at a gray 256 level. The photosensor can provide the right quality image under weak optics and slow velocity in a period of TDI running. The photosensor grabs an image of a moving object during transfers of the charge in synchronous scanning with the object. 

On the other hand, the image process usually transforms from the full color, gray level, binary threshold to black and white to simplify the intricate image. The task carries on the gray level as fast and accurate as experimentally, in which the gray level from 0 to 255 variety illustrates its range. Moreover, there are three phases to process the image after scanning, and the first step is image acquisition. The image is acquired by a line sensor, which is along with an optical ruler using gantry. The second step processes the image feedback to the computer to detect the image by subtraction. The defect pixel is extracted on the image by an automatic data acquired program (ADAP). Then the defect pixel is merged and identified its size and gray level. The gray level is from 0~255 with a 256 gray variety, in which 0 is the blackest, and 255 is the whitest. TFT defects are clustering and classification types at post-processing, such as open or short, or other types of defects. Furthermore, the background is essential to the image separation out foreground samples from the background at the continuous image. Generally, the background subtraction can detect the moving objects from line-scan CCD then takes the image process to mark the defect. 

Figure 2 illustrates the flow process of images extracted by automatic optical inspection. The image calculates its gray level intensity and marks the defect position to be judged the defect sizing using the binary process by the ADAP. At first, the image acquired by line-scan, which the conversion is from analog to digital, and the image sent to computer calculation about the defect pixel extraction, merge pixel defects sizing. Then, the image is post-processing clustering, classification, and report the defect in the display. Furthermore, Figure 3A indicates the sample for three defects of the morphology from top to bottom: a black square, gray oval, and white ellipse. The gray intensity is 0, 125, and 255, accordingly. Figure 3B indicates the background for three defects of the morphology from top to bottom as gray square, black oval, and gray ellipse. Figure 3C is the subtraction between Figure 3A and Figure 3B. Moreover, Figure 3D is an indication of the grayscale. 

In contrast, the gray intensity is 125, 0, and 125, accordingly. Figure 3A,B indicates the background subtraction for three defects of the morphology from top to bottom as gray square, gray oval, and gray ellipse. In contrast, the gray intensity is −125, 125, and 130, respectively.

At first, the task implements low-pass filtering to reduce the pixel’s noise. The task carries out a linearization of the gray value to distribute the pixel’s image and strengthen the contrast between white and black. The gray value distributed in the image of the pixel is modified. The extension process enhances the contrast to help the row data to distinguish the blemish from the background. Moreover, the binarization is a threshold to the basis of the gray level on the image. In the experiment, the grayscale of an image is divided into only two kinds of gray values; that is, a gray value is set. Suppose the gray of the image is greater than the threshold that is a bright spot. If the gray value is lower than the set value, that is a dark spot. In this way, a binary image can be obtained. First, a complex picture to simplify, the reticle cutting image, is commonly used to detect smooth surface workpieces’ flaws. It is assuming that m is the threshold value of binarization. Equation (1) sets the image grayscale value m, as below [21,22].
(1)$$m\;=\;\overset{n}{\underset{i=1}{\sum }}f\left(x,y\right)$$

Whereas: *f* is the input image, n is the number of all pixels, *f*(*x*,*y*) indicates the gray value of pixel coordinates (*x*,*y*) above the equation. If the image’s gray value is lower than the grayscale value m, let it be 0, and if the gray value of the image is higher than the division value m, let it be 1. This technique is called a binary threshold. The task selects the appropriate binarization threshold m so that *f*(*x*,*y*) > *m* then *f*(*x*,*y*) is set to 255; *f*(*x*,*y*) < *m* as the *f*(*x*,*y*) is set to 0, this is the binary theory in the image process.

## 3. Experiment Architecture

The experiment is carried out in the clearing room at 10 class levels [23,24] using the glass 8.5th generation, dimension size 2500 × 2200 mm, manufactured by Corning Inc. (Corning, NY, USA). The room temperature and the experimental humility condition follow the Federal Standard 209E [25,26]. The OS sensor specification utilizes the 88,000 ATS-620 Array Test system, fabricated by Keysight (Santa Rosa, CA, USA). That is a fast TFT array tester of a-Si LCD panels, and the defect detection capability with a short time is at 3 ms. Additionally, the test platform linked the pneumatic caution prevents the antivibration from four foundations. That is designated for granite material with planarity deviation at 1 μm, in which the thickness is 200 mm. Moreover, the TDI-CCD line scan optical resolution is at 1 μm and even lower (0.5 μm) if the magnified lens is installed on the photosensor.

Figure 4 illustrates the methodology of an open/shore sensor unit engaged with the automatic optical inspection. The glass-in of the section is in Figure 4A, the open/short sensor and probe attached with the TFT panel are in Figure 4B, and automatic optical inspection of TDI-CCD linked with a light emitted diode (LED) and the lens is ready to scan the Figure 4C. Moreover, the area-CCD with five lenses is ready to review the defect after line scanning in Figure 4D, which assumes 60 defect points on the 8.5 generation glass that takes one defect for one second, and the sample finishes the test delivery out in Figure 4E. The architecture consists of the five sections of Figure 4A–E, respectively. At first, Figure 4A is the conveyor to allow glass input to align the right position. Section Figure 4B installed the OS sensor and probe to detect the metal electrical conductivity while the substrate enters the device by the conveyor. The substrate is then delivered to section Figure 4C vacuum planarity using air suction for the optical line-scan. After that, the substrate transfers to section Figure 4D for the defect reviewed by area-CCD. The review takes the 2×, 5×, 10×, 20×, and 50× of the lens magnification with a color image. Section Figure 4E is the glass-out, then the sample to the next station is for further manufacture. The 8.5 generation glass cycle time takes approximately 180 s to test the electrical conductivity throughout the panel from right to left and top to down, another 60 s for the TDI-CCD line scan, and a 60 s for review defect by 10 μm resolution. The sequence is the glass-in with alignment, detect the electrical conduction, TDI-CCD line scan, review CCD of the defect, and the glass-out. Thus the total takes a cycle time of 320 s. Whereas Table 1 displays a cycle time chart as below.

On the other hand, the experiment takes a diagonal 7-inch quartz mask, an opaque plate with a defined defect pattern, manufactured by JD Photo data Inc (Hitchin, Herts, UK), which is used to verify the capability of the optical resolution. Figure 5 illustrates the left pattern of the photomask for further investigation. The photomasks with the consistency of Figure 5A–D correspond to the metal layer, normal area, defect area, and test key, respectively, which provides the various patterns on the surface of the quartz. Figure 5A is a metal property to check the electrical connections, Figure 5B is the normal area, and Figure 5C indicates four kinds of defects to check the photosensor capability, and Figure 5D indicates a test key to check the scanning ability. Figure 6A–D demonstrates the pattern of the photomask for another investigation. The sections are consistent with Figure 6A defect size, Figure 6B defect type, Figure 6C inside the 7-inch mask, and Figure 6D outside the 7-inch mask.

In the paper, the task investigates the Figure 6D of the right pattern to verify the TDI-CCD scanning ability. Because TDI-CCD can detect a micro defect, several tailor-made kinds of defect patterns are on the mask. However, the pixel circuit with various resolutions in TFT is dependent on the product requirement. Anyone defects smaller than ten micro dimensions adhered to the TFT incurs low electricity conductively. Therefore, the AOI can be the keeper of the zero-defect in the TFT industry.

## 4. Results and Discussion

The task utilizes the OS sensor unit engaged with AOI and various magnified lenses to detect the glass 8.5th generation. As a result, Figure 7 presents the open defect on the metal line to be open-circuit at sensor no.4 as the red mark; thus, a tailor-made demonstration of the open defect on TFT illustrates the Figure 7A–C. The metal line of Figure 7A is detected in an open circuit by the OS sensor so that the *E_out_* voltage of Figure 7B cannot measure while the condition is *E_in_* = 100 V at 60 Hz and forward current at 1A. As the resistor is approaching the infinite, the circuit does not close the loop. Figure 7C indicates the interruption of the power on a metal line open circuit rather than a normal line 100 W. As a result, its indication is no current flow to the metal line at the open condition, which the sensor receives a low output value at the location of the defect.

Figure 8 demonstrates the short defect as the red mark on the metal line to the short-circuit; thus an illustration of the short defect on TFT schemes the Figure 8A–C. Figure 8A detected a short circuit by OS sensor so that the *E_out_* voltage Figure 8B measures *E_in_* = 100 V at 60 Hz and forward current at 1A, since the resistor measures at 100 Ω, whereas the circuit is getting a short loop. Figure 8C indicates the normal parallel line at the power of 100 W. As a result, each sensor receives twice the average output power of 200 W, so the value is two times higher at the short defect condition. Thus, the short defect on TFT cannot be detected because the OS sensor receives a specific value than the normal condition.

Figure 9 presents the cross-short (X-short) defect on the metal line at sensor no. 4 as the red mark to the scheme in the Figure 9A–C. The Figure 9A detected an X-short circuit by OS sensor. Thus, the *E_out_* voltage Figure 9B measures *E_in_* = 100 V at 60 Hz and forward current at 1A, since the resistor measures at 100 Ω, whereas the circuit is getting a short loop. The Figure 9C of indicates the regular parallel line at the power (W). As a result, the sensor detects the X-short defect at a pick at sensor no. 4 of 200 W power compared to the average output power of 100 W, so the value is high. At the same time, the other sensors are normal power.

The automatic optical inspection is based on the line-scan to grab the image and the axial direction by the TDI-CCD photosensor installed on the vision machine, which is the sharpest optical sensor among the line-scan devices. The TDI-CCD is the most multi-scan to simultaneously take one image and accumulate the multiple exposures that save time and enhance image quality. At the same time, the samples under test move forward efficiently and effectively. The experiment utilizes the existing photomask to check the scan capability.

Figure 10 illustrates the result by line-scan, in which the defect is on the mask, such as the short and open on the surface. Figure 11 presents another pattern of the 7-inch photomask to detect its existing open and short defect on the surface. As a result, the detectability is quite sharply and preciously scanned by TDI-CCD.

On the other hand, the experiment also utilizes the color area-CCD to magnify TFT defect for further inspection. The AOI installed a set of area-CCD at the diagonal length of 1/2 inches to take the image of the dimensional area for the field of view (FOV). The lens engaged with the various magnification for 2×, 5×, 10×, 20×, and 50×, respectively. Figure 12 indicates several images concerning the magnification at the upper right corner. Moreover, higher magnification results from the image quality due to vibration, whereas antivibration is another issue for AOI. The AOI platform installed the pneumatic caution to prevent the vibration, which the cause is the most by an unstable foundation or resonance equipment in the factory.

Table 2 addresses several lenses, a set of 350,000-pixel area-CCD, pixel number, resolution, and field of view (FOV) with various magnifications. The 50× magnification is a critical saturation as the smallest FOV 128 × 96 μm on the presented task. Thus, the environmental condition requests a planarity stage to prevent out of focus. The task utilizes a set of objective turrets installed five sets of the lens in the experiment, which descending order are 2×, 5×, 10×, 20×, and 50×, respectively.

## 5. Conclusions

The task investigates the open and short defect characteristics on a thin-film transistor array using an electrical tester engaged in automatic optical inspection. The sensor does not detect the current flow at the open defect condition, and the electrical conductively is poor. At the short defect, the sensor receives twice as normal as the metal line at the power. In a short cross condition, the senor receives a vigorous-intensity at the power than normal metal line. Furthermore, the quartz mask installed on AOI verified its optical resolution; the TDI-CCD can grab an image of a moving object during transfers of the charge in synchronous scanning with the object. Consequently, the open/short sensor engaged with AOI plays an inspector to free defects on the TFT is very impressive.

## Figures and Tables

**Figure 1 micromachines-12-00135-f001:**
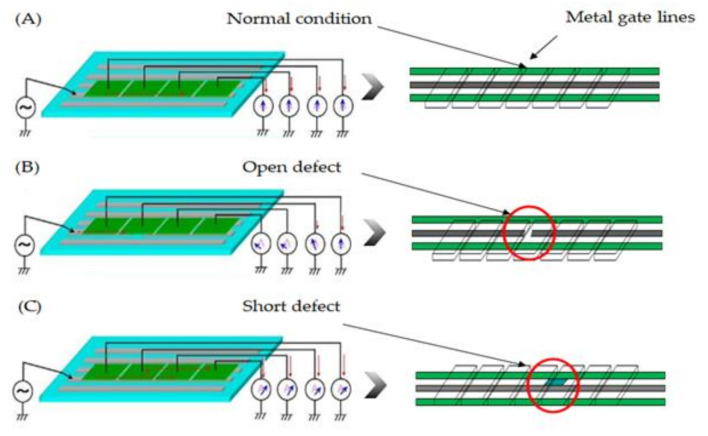
A topology of open/short defects appearing on the thin-film transistors (TFT) display, (**A**) normal condition, (**B**) open defect, and (**C**) short defect.

**Figure 2 micromachines-12-00135-f002:**
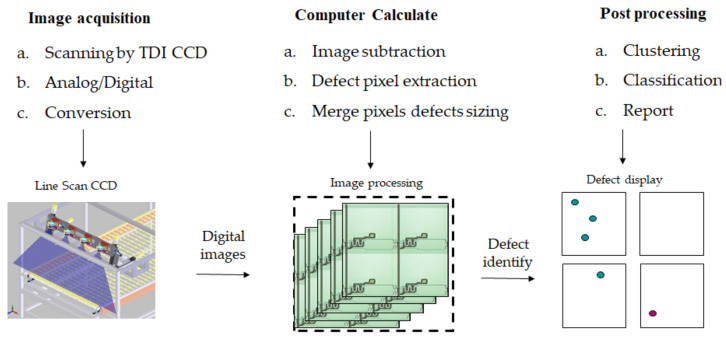
A process of an image extracted by automatic optical inspection.

**Figure 3 micromachines-12-00135-f003:**
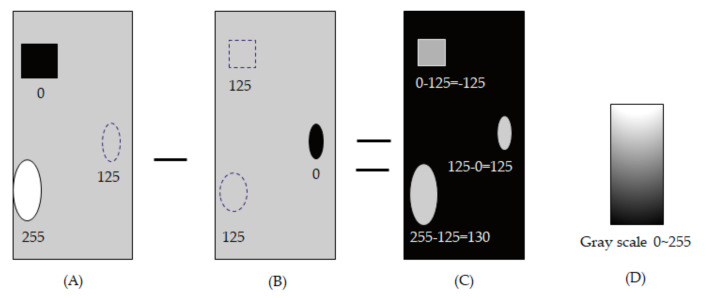
A process of the background subtraction for gray level value, (**A**) shown three defects, (**B**) the background, (**C**) the subtraction, and (**D**) indicator of grayscale.

**Figure 4 micromachines-12-00135-f004:**
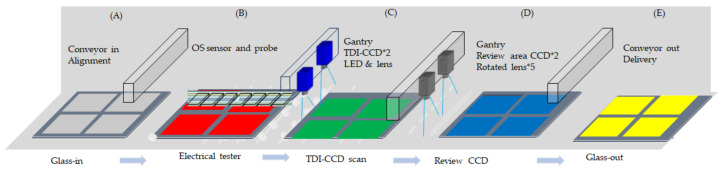
An architecture of open/short sensor unit engaged with the automatic optical inspection, (**A**) glass-in, (**B**) OS sensor unit, (**C**) scan TDI-CCD, (**D**) review CCD, and (**E**) glass-out.

**Figure 5 micromachines-12-00135-f005:**
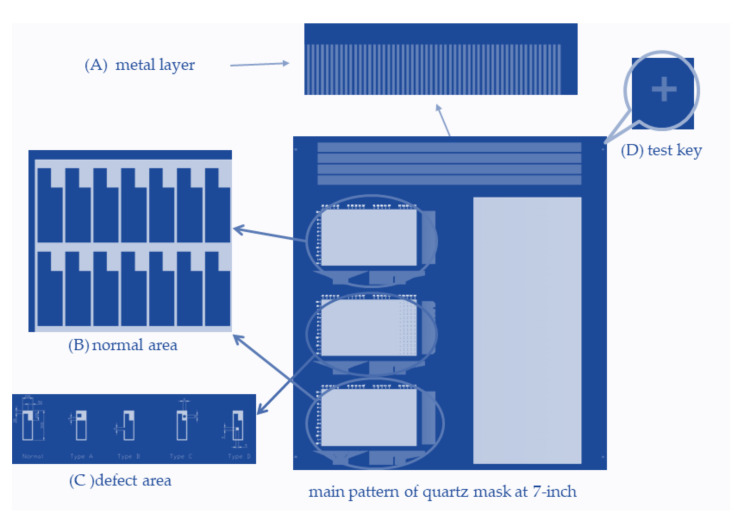
The right patter at (**A**) metal layer and (**D**) test key, and the left pattern at (**B**) and (**C**) of the 7-inch mask scheme for AOI verification.

**Figure 6 micromachines-12-00135-f006:**
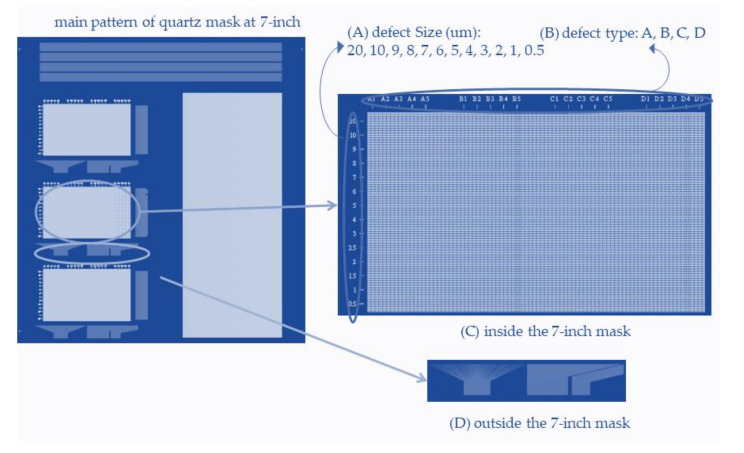
The pattern at (**A**–**D**) of the 7-inch mask scheme for automatic optical inspection (AOI) verification.

**Figure 7 micromachines-12-00135-f007:**
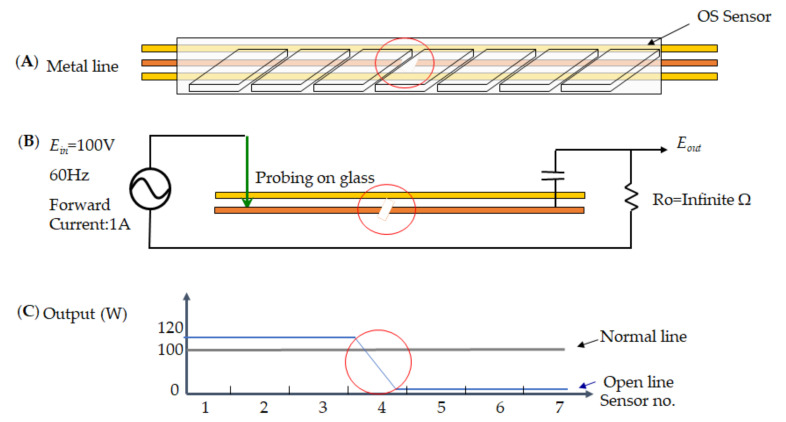
An open defect presents on the electrical signal for thin-film-transistor (TFT), which (**A**) metal line, (**B**) measurement voltage, and (**C**) power distribution.

**Figure 8 micromachines-12-00135-f008:**
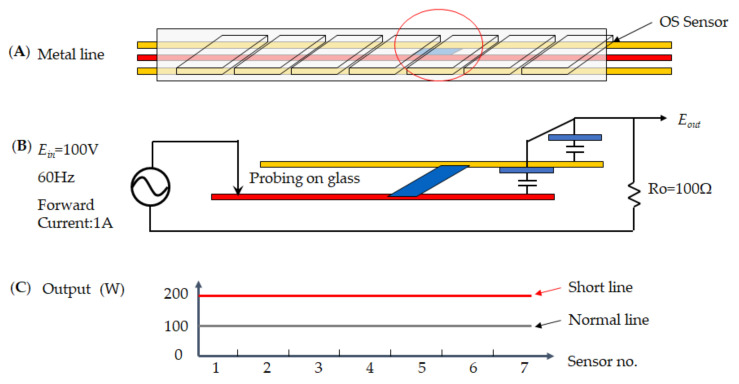
A short defect illustrates the electrical signal for TFT, which (**A**) metal line, (**B**) measurement voltage, and (**C**) power distribution.

**Figure 9 micromachines-12-00135-f009:**
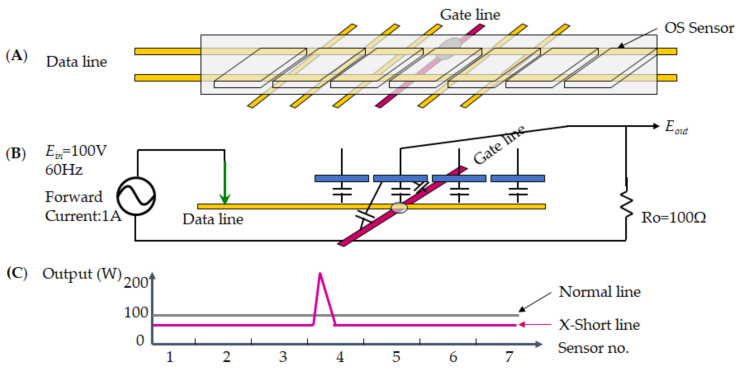
An X-short defect presents on the electrical signal for TFT, which (**A**) metal line, (**B**) measurement voltage, and (**C**) power distribution.

**Figure 10 micromachines-12-00135-f010:**
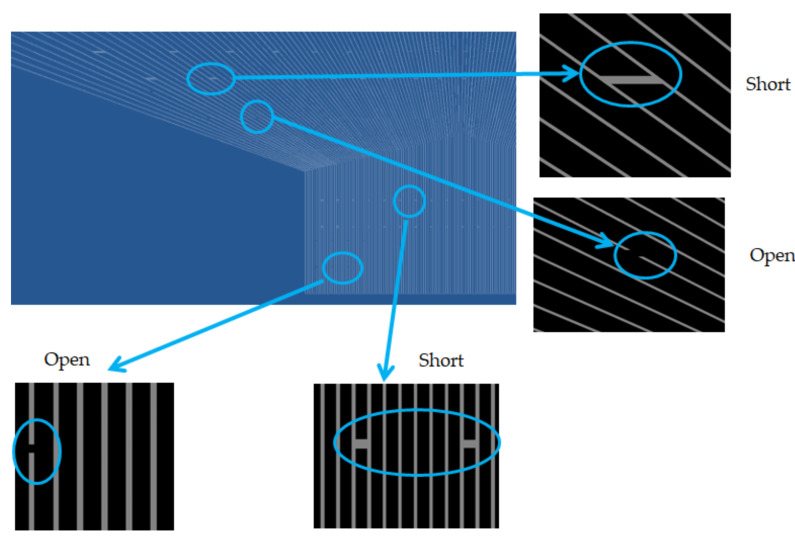
Verify at open and short defect on the right area of the photomask for AOI.

**Figure 11 micromachines-12-00135-f011:**
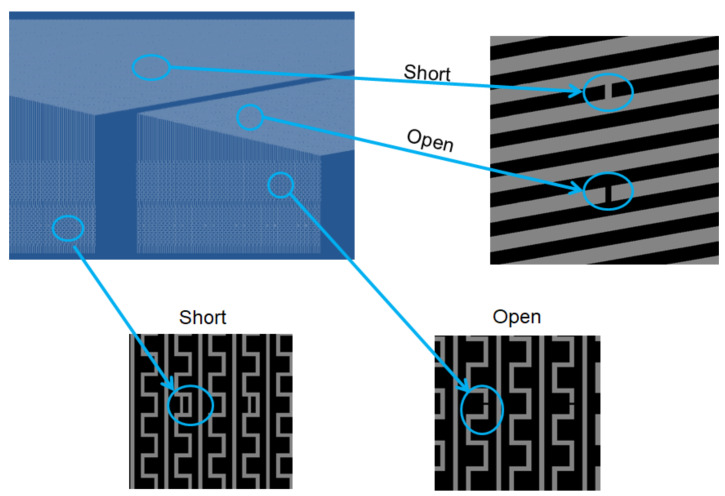
Verify at open and short defect on the left area of the photomask for AOI.

**Figure 12 micromachines-12-00135-f012:**
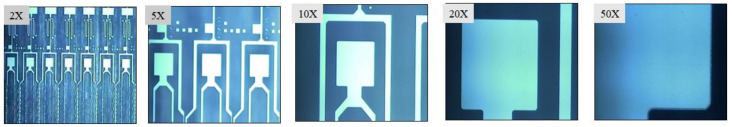
Various magnification lens with 2×, 5×, 10×, 20×, and 50× are illustrated by area-CCD.

**Table 1 micromachines-12-00135-t001:** The 8.5 generation glass cycle time illustrates the time chart.

Sequence	Glass-in	Electrical Tester	TDI-CCD	Review CCD	Glass-out	Total
Time	10	180	60	60	10	320

Unit: second (s).

**Table 2 micromachines-12-00135-t002:** The area-CCD illustrates the magnification, resolution, and field of view (FOV) by various lenses.

Lens	CCD	Pixel Number		Resolution		FOV	
Magnification	Diagonal	W	H	W (μm)	H (μm)	W (μm)	H (μm)
2×	1/2”	640	480	5	5	3200	2400
5×	1/2”	640	480	2	2	1280	960
10×	1/2”	640	480	1	1	640	480
20×	1/2”	640	480	0.5	0.5	320	240
50×	1/2”	640	480	0.2	0.2	128	96

Note: W (Width), H (Hight).

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
