# Peer review of "Effectiveness of Electrical and Optical Detection at Pixel Circuit on Thin-Film Transistors"

_micromachines, 2021, doi:10.3390/mi12020135_

Round 1
Reviewer 1 Report
Methodology is desribed well.
But the experimental results is not sufficient.
Please show the scale of output (kW) in Fig. 7(C), 8(C), 9(C).
Please decribe the test time for the experiment using 8.5 G glass.
The objective of this study is the methodology of defect detection in circuits.
So I think that the lines 25-30, and lines 38-52 describing general TFT structures and the trend of FPD market should be written more compactly.
Author Response
Point-by-point reply to Reviewers' comments and suggestions
Dear Editor,
The author is the great appreciation of reviewers' comments and suggestions for the paper. The paper has been revised accordingly. Those for Reviewer 1 are marked in red, and those for Reviewer 2 in blue wise for ready reference. Details of the point-by-point replies are as follows.
Sincerely,
Fu-Ming Tzu

Reviewer 2 Report
Comments to the Author
In this manuscript, the authors reported the effectiveness of electrical and optical detection on thin-film transistors (TFT). The presentation of the paper is quite impressive. The physical characterizations, electrical and optical characteristics are quite relevant. This work seems very interesting and organized. Although, there are some technical errors that need to be improved. I think the manuscript should be published after minor revision. Some minor comments and suggestions are listed below to the authors:
- The English writing of the manuscript should be improved. There are many grammatical mistakes all over the manuscript. In this case, authors may seek help from professionals.
- There are many typing errors in the manuscript, which need to be addressed. Such as, in line 186 it is written ‘low’ which should be ‘Low’. and another mistake is in line 213 and it is written ‘shore’ which should be ‘short’. In line 241 it is written ‘Ein’ which should be ‘Ein’.
- In the figures, subscripts and superscripts are not properly written. Such as in 7, it is written ‘Ein’ which should be ‘Ein’, and similarly, Rout should be Rout and so on.
- In 7 and in Fig. 8, it is written ‘grass’ which should be ‘glass’.
The authors are encouraged to address the above-mentioned points.
Author Response
Point-by-point reply to Reviewers' comments and suggestions
Dear Editor,
The author is the great appreciation of reviewers' comments and suggestions for the paper. The paper has been revised accordingly. Those for Reviewer 1 are marked in red, and those for Reviewer 2 in blue wise for ready reference. Details of the point-by-point replies are as follows.
Sincerely,
Fu-Ming Tzu
Reply to Reviewer 1 Comments
Response to Reviewer 2 Comments
Point 1: The English writing of the manuscript should be improved. There are many grammatical mistakes all over the manuscript. In this case, authors may seek help from professionals.
Response 1: Thanks for the reviewer's advice. The paper consults the expertise to amend and edit the writing structure. The author also takes professional software, Grammarly Premium, checks the content throughout the paper, and can be a ready reference.
Point 2: There are many typing errors in the manuscript, which need to be addressed. Such as, in line 186 it is written 'low' which should be 'Low'. and another mistake is in line 213 and it is written 'shore' which should be 'short'. In line 241 it is written 'Ein' which should be 'Ein'.
Response 2: Revised accordingly. As below,
- Many thanks for the reviewer's comment. The author finds the 『low 』is a typo; that is the author's fault. The meaning correct to 『A 』. Thus, 『A process of an image extracted by automatic optical inspection』in figure 2.
- Figure 4. An architecture of open/short sensor unit engaged with the automatic optical inspection.
- The condition is Ein=100V at 60HZ
Point 3: In the figures, subscripts and superscripts are not properly written. Such as in 7, it is written 'Ein' which should be 'Ein', and similarly, Rout should be Rout and so on.
Response 3: Thanks for the reviewer comment; the revision is accordingly.
Fig. 7 An open defect presents on the electrical signal for TFT
Fig. 8 A short defect illustrates the electrical signal for TFT
Fig. 9 An X-short defect presents on the electrical signal for TFT
Point 4: In 7 and in Fig. 8, it is written 'grass' which should be 'glass'.
Response 4: Thanks for the reviewer comment; the revision is accordingly.
Fig. 7 An open defect presents on the electrical signal for TFT
Fig. 8 A short defect illustrates the electrical signal for TFT
Round 2
Reviewer 1 Report
The manuscript has been improved.